# Socio-Ecological Natural Experiment with Randomized Controlled Trial to Promote Active Commuting to Work: Process Evaluation, Behavioral Impacts, and Changes in the Use and Quality of Walking and Cycling Paths

**DOI:** 10.3390/ijerph16091661

**Published:** 2019-05-13

**Authors:** Minna Aittasalo, Johanna Tiilikainen, Kari Tokola, Jaana Suni, Harri Sievänen, Henri Vähä-Ypyä, Tommi Vasankari, Timo Seimelä, Pasi Metsäpuro, Charlie Foster, Sylvia Titze

**Affiliations:** 1UKK Institute for Health Promotion Research, P.O. Box 30, 33501 Tampere, Finland; johanna.tiilikainen@tuni.fi (J.T.); kari.tokola@ukkinstituutti.fi (K.T.); jaana.suni@ukkinstituutti.fi (J.S.); harri.sievanen@ukkinstituutti.fi (H.S.); henri.vaha-ypya@ukkinstituutti.fi (H.V.-Y.); tommi.vasankari@ukkinstituutti.fi (T.V.); 2Department of Transport and Streets, City of Tampere, Frenckellinaukio 2, PL 487, 33101 Tampere, Finland; timo.seimela@tampere.fi; 3Department of Mobility and Transport, WSP Finland Ltd., Kelloportinkatu 1 D, 33100 Tampere, Finland; pasi.metsapuro@wsp.com; 4Centre for Exercise Nutrition and Health Sciences, School for Policy Studies, Faculty of Social Sciences and Law, University of Bristol, 8 Priory Road, Bristol BS81TZ, UK; Charlie.foster@bristol.ac.uk; 5Institute of Sport Science, University of Graz, Mozartgasse 14, 8010 Graz, Austria; Sylvia.titze@uni-graz.at

**Keywords:** active travel, workplace, natural experiment, multilevel, intervention

## Abstract

Active commuting to work (ACW) has beneficial effects on health, traffic, and climate. However, more robust evidence is needed on how to promote ACW. This paper reports the findings of a multilevel natural experiment with a randomized controlled trial in 16 Finnish workplaces. In Phase 1, 11 workplaces (1823 employees) from Area 1 were exposed to environmental improvements in walking and cycling paths. In Phase 2, five more workplaces (826 employees) were recruited from Area 2 and all workplaces were randomized into experimental group (EXP) promoting ACW with social and behavioral strategies and comparison group (COM) participating only in data collection. Process and impact evaluation with questionnaires, travel diaries, accelerometers, traffic calculations, and auditing were conducted. Statistics included Wilcoxon Signed Ranks Test, Mann-Whitney *U*-test, and after-before differences with 95% confidence intervals (95% CI). After Phase 1, positive change was seen in the self-reported number of days, which the employees intended to cycle part of their journey to work in the following week (*p* = 0.001). After Phase 2, intervention effect was observed in the proportion of employees, who reported willingness to increase walking (8.7%; 95% CI 1.8 to 15.6) and cycling (5.5%; 2.2 to 8.8) and opportunity to cycle part of their journey to work (5.9%; 2.1 to 9.7). To conclude, the intervention facilitated employees’ motivation for ACW, which is the first step towards behavior change.

## 1. Introduction

Regular physical activity enhances physical and mental health and improves quality of life [1]. Active commuting to work (ACW) by walking or cycling increases the total amount of physical activity [2,3] thereby decreasing health risks [4] and costs [5]. Active travel in general may also reduce carbon emissions [6] and air pollution [7], which are amongst the leading environmental health risks globally [8] causing detrimental cardiovascular effects [9] and premature deaths [10].

Despite its indisputable benefits, the share of walking and cycling from all daily trips is 25 to 35% in most European countries and only around 10% in Australia, Canada, and US [11]. However, the data collection methods vary greatly from country to country, which has led to attempts to harmonize national travel statistics in Europe [12]. The results show that based on kilometers per person, the modal split of walking and cycling is less than 10% in many countries. Work presents one of the most common purposes for traveling [12], but for example in England [13] and Finland [14], less than 20% of work trips is primarily walked or cycled.

It is therefore understandable that the decision makers and practitioners in the fields of health, environment, and traffic constantly seek efficient ways to increase the proportion of walking and cycling and other sustainable transport modes in work trips and in general. The recently released statements on global warming have made the need even more urgent and for instance in Finland the goal is to increase the modal split of walking and cycling by year 2030 from the current 30 to 35–38% [15].

Adversely, based on the recent systematic reviews conducted in the field of physical activity, health psychology, and transportation, the evidence base for changing travel behavior is still developing [16,17,18] and can currently provide only limited information on the effectiveness of intervention designs and behavior change techniques to promote walking and cycling [18]. Conceptual frameworks have been reviewed to gain understanding of the relevant determinants, confounders and mediators of active travel behavior and synthesized to develop more comprehensive frameworks for active travel research (e.g., PASTA project) [19]. As a result, it is generally acknowledged that behavioral choice particularly about ACW is not based only on physical environment and psychological factors but also on social settings and obligations [20,21]. This makes commuting “a complex social phenomenon and practice” [22] advocating need for interventions that target actions at multiple levels [23].

However, to date, studies promoting ACW have primarily focused on individuals and multilevel approaches targeting also at interpersonal, organizational, and infrastructural factors have seldom been utilize [16,19]. This paper reports the findings of such study by also embedding a randomized controlled design within the context of natural experiment in a Finnish worksite setting. This is a unique and highly appreciated approach in the field of active travel research, where a need for longitudinal, large scale interventions conducted in real-world settings has been identified [23]. The findings include process evaluation, behavioral impacts and the use and quality of the main walking and cycling path. The study protocol has been previously reported [24].

## 2. Materials and Methods

A brief summary of the study materials and methods is provided here. More detailed description can be found from the protocol article [24].

### 2.1. Ethics

The study was approved by the Ethics Committee of the Tampere Region (running number 20/2014). The participants gave their consent to participate in the study by agreeing to complete the measurements after being fully informed in writing about the ethical principles and data protection. Written informed consent was not obtained because no sensitive data were collected, and the study did not intervene physical integrity of the participants. The study has been registered at ClinicalTrials.gov (NCT01633918) and the CONSORT checklist extended for cluster randomized controlled trials is presented in Appendix A.

### 2.2. Participants

Small and middle-sized companies with more than 10 employees were recruited from two large business areas (Area 1 and Area 2) located just outside the city center of Tampere, Finland. After the workplace enrolment the employees were invited to participate in the study via email. Prior to baseline measurements a short kick-off meeting was organized in each participating workplace.

### 2.3. Intervention

The intervention was based on socio-ecological approach [25,26]. It is a systems model that emphasizes health; consequently, health behavior is affected by dynamic interplay between intrapersonal factors, as well as physical and social environments. Actions are therefore needed at multiple levels to change human behavior. In this study, the actions were first targeted at the physical environment (Phase 1) and then at the individual employees and their working units and worksites (Phase 2). The graphical illustration of the approach has been presented in the protocol article [24].

Phase 1 involved 11 workplaces from Area 1. It assessed the effects of environmental improvements on employees’ ACW in a longitudinal pre-post intervention design. The improvements were made to the main and connecting walking and cycling paths and were undertaken by the city of Tampere using regional funding of land use, housing, and transportation.

Phase 2 assessed the effectiveness of workplace-specific social and behavioral strategies on employees’ ACW in a cluster-randomized controlled condition. For this purpose, five more workplaces were enrolled from Area 2 to increase the statistical power. The areas were similar in terms of how the workplaces were situated along the main walking and cycling path. Due to the environmental improvements to Area 1 in Phase 1, the quality of the paths was also comparable between the areas in Phase 2 [24]. After the additional recruitment the workplaces in Area 1 and Area 2 were arranged into pairs according to the number, sex distribution and educational level of the employees’ as well as to the proportion of employees’, who had a possibility to walk or cycle to work but did not report to do so in the baseline questionnaire. These workplace-pairs were then randomly assigned into either an experimental (EXP) or comparison group (COM).

Each workplace in EXP nominated an internal team, which carried out social and behavioral strategies to promote ACW. The researchers supported the teams with a workbook of social and behavioral strategies categorized into individual, working unit, and organization levels, other free of charge material and three visits. The workplaces in COM participated only in data collection but were offered the possibility to receive the same support after the study.

### 2.4. Evaluation

In Phase 1, the baseline measurements were conducted in autumn 2014 and Spring 2015 (M1) and the follow-up measurements after environmental improvements in Fall 2016 (M2). In Phase 2, the information from M2 served also as a baseline in Area 1, whereas in Area 2, completely new baseline measurements were needed in the newly recruited workplaces. The follow-up measurements after social and behavioral strategies took place in both areas in Spring 2017 (M3). Throughout the study, all measurements were carried out either in spring or fall and were seasonally the same in each workplace.

The indicators and measures of process and impact evaluation are described in Table 1. The questions in the employee questionnaire concerning impact evaluation are presented in the Appendix A. For a more objective evaluation of ACW, the employees were also asked to wear a tri-axial accelerometer [27,28] and to use a travel diary (File S1), preferably for seven days. Employees, who used accelerometers, were provided graphical feedbacks about their physical activity and sedentary behavior. In Phase 1, traffic calculations in four counting points during afternoon peak hour and auditing by bicycle were used to examine the changes in the general use (average number of pedestrians and cyclists trespassing the main path) and quality of the main walking and cycling path (speed and comfort of cycling, separation of walking, and cycling paths) after environmental improvements [24].

### 2.5. Statistics

Statistical power calculations for Phase 2 were based on independent samples t-test in a cluster-randomized design. The standard deviation was assumed to be 3.0 and the intra-cluster correlation 0.05. To detect a between-group difference of one day in the weekly change of walking or cycling to work, eight workplaces per group and 64 employees per workplace were needed to achieve 80% power.

In Phase 1, Wilcoxon Signed Ranks Test was used for continuous variables to test the effect of environmental improvements. For nominal variables, difference of after-before percentages and a 95% Wald interval for a difference of proportions with matched pairs were calculated. To show the effect of social and behavioral strategies in Phase 2, Mann-Whitney *U*-test was used to test the between-group difference of change in the continuous after-before variables. For nominal variables difference of two independent percentages with a 95% confidence interval were calculated.

## 3. Results

### 3.1. Process Evaluation (Phase 1 and 2)

#### 3.1.1. What Percentage of Workplaces Volunteered and How Representative Were They?

In Phase 1, 37 companies with initial interest were approached and 11 (30%) of them agreed to participate. In Phase 2, seven more workplaces were approached from Area 2 and five of them agreed to participate. Thus, 16 out of 44 (36%) workplaces participated representing infrastructural services, engineering, industrial technology, information technology, telecommunications, software development, public services, health and social services, heavy industry, and education [24]. The workplaces participating in the study covered the main fields of activity in both areas except for car retailers in Area 1, who were not contacted due to their mild preliminary interest.

#### 3.1.2. What Percentage of Potentially Eligible Employees Took Part and How Representative Were They?

Phase 1 included 11 workplaces with 1823 employees. Of those employees, 900 (49.4%) of them responded to the baseline questionnaire and thus participated in the study (Figure 1).

Only a limited amount of information was available regarding the working-age population in the Tampere region and more specifically in the study area. Respondents’ demographics (Statistics Finland) and average length of the work trip (Finnish Transport Agency) were comparable to the previous data collected from the general adult population in Tampere region (Table 2). However, the respondents more often had a university degree, which may indicate that more educated employees responded to the questionnaire in Phase 1 and participated in the study. Alternatively, the employees’ educational level may have been higher in the study area than on average in the Tampere region.

Phase 2 included 16 workplaces from Area 1 and Area 2, accounting altogether 1228 employees (Figure 2). Baseline information was obtained from 630 (51.3%) employees. Their background was similar to Phase 1 with the exception that the respondents in Phase 2 had even more participants with a university degree (Table 2).

#### 3.1.3. To What Extent Did the Strategies Succeed as Intended?

In Phase 1, the improvements of the main walking and cycling path within the area were planned to be ready in Fall 2015. However, they were delayed by one year and completed in Fall 2016. As the follow-up measurements (M2) were due in the same fall, the exposure time to environmental improvements shrank from six months to the maximum of two months, depending on the workplace.

In Phase 2, the internal teams of the eight workplaces in EXP started to make their action plans for social and behavioral strategies in Fall 2016 (after M2). They implemented the strategies from September 2016 to May 2017 before the last measurements (M3). All workplaces returned their action plans to the researchers. The plans included a selection of 22 strategies (Appendix A). Of those, 10 focused on organization, 7 on working unit, and 5 on individual level. The number of strategies per workplace varied from 2 to 16 (10 on average). Of those, six workplaces implemented strategies at all three levels and two at two levels.

#### 3.1.4. To What Extent did the Employees Acknowledge the Strategies Implemented?

After the environmental improvements in Phase 1 (M2), the employees were asked whether they had made any preparations to change their ACW and whether the improvements had affected their ACW (perceived effects). Almost one half of the respondents (46.3%) reported having made some preparations for ACW: 17% had repaired old bicycle or bought a new one, 16% had sought suitable routes, 15% had purchased new equipment for walking or cycling, and 14% had considered ways to include some walking or cycling in their journeys. However, almost all respondents perceived that the improvements had not affected their ACW behavior to walk and cycle to work (99% and 95%, respectively).

After social and behavioral strategies in Phase 2 (M3) the employees were asked whether they had acknowledged actions to promote ACW in their workplaces. Of those responding to the question (*n* = 378), 55% reported that they had noticed some actions and 26% that they had taken part of them.

### 3.2. Behavioral Impacts (Phase 1 and 2)

#### 3.2.1. Motivation for ACW (Willingness, Opportunities, and Intention)

Before the environmental improvements in Phase 1, 43% of the participants reported willingness to increase walking and 62% willingness to increase cycling to work (Table 3). Correspondingly, 61% of the participants considered having an opportunity to walk and 65% to bicycle at least part of their work trips. However, less than 2% of the participants intended to do so in the following week. After environmental improvements, a statistically significant change was discovered in the self-reported mean number of days with intention to bicycle at least part of the work journey in the following week (*p* = 0.001) (Table 4). 

Before social and behavioral strategies in Phase 2, the proportion of participants, who were willing to increase walking and bicycling to work and who reported having opportunities to do so were somewhat lower in EXP than COM (Table 5). Similarly, the mean number of days with intention to walk or cycle to work in the following week was slightly lower in EXP than COM.

After social and behavioral strategies, a statistically significant positive effect was discovered in EXP in the proportion of participants, who reported willingness to increase walking (8.7%; 95% CI 1.8 to 15.6) and bicycling (5.5%; 95 %CI 2.2 to 8.8) and in the proportion of participants, who reported having an opportunity to bicycle at least part of their work journey (5.9%; 95% CI 2.1 to 9.7) (Table 4).

#### 3.2.2. Employees’ ACW (Primary Mean of Transportation, Weekly Number of Days Actively Commuting to Work and From Work)

Baseline information from Phase 1 shows that cars were employees’ primary mean of transportation to and from work (Table 5). The employees walked part of their journey to work on average one day a week with the mean of 0.8 km and eight minutes. The weekly number of days was the same in cycling with the mean of 2.5 km and nine minutes.

In the follow-up measurements, 10 workplaces were still involved (M2). One workplace employing 28 persons had moved from the area and in the remaining workplaces the number of employees had reduced by 150 from those responding to the baseline questionnaire (*n* = 900) (Figure 1). As a result, the total number employees invited to M2 was 722 (58.8% of the original sample). Of them, 402 (55.7%) responded to the follow-up questionnaire and 246 (34.1%) had questionnaire data from both measurement points. After environmental improvements, no self-reported changes were discovered in any of the ACW variables (Table 6). Changes in more objectively measured ACW were not analyzed as only 43 (6.0%) employees had accelerometer and diary data from both measurement points (Figure 1).

In the beginning of Phase 2, there were slightly less participants in EXP than in COM, who belonged to the youngest age-group (10% vs. 20%) (Table 2). The participants in EXP did also more often regular day work and sedentary work than the participants in COM (90% vs. 82% and 94% vs. 79%, respectively). Some baseline differences between the groups were also seen in self-reported ACW (Table 3): Participants in EXP used cars more often as a primary transportation mode to work and from work (58% and 58%) than the participants in COM (49% and 49%). Consequently, going by foot and bicycle to work and from work was less common in EXP (to: 8.6% and 17.4%, from: 8.6%, 17.6%) than in COM (to: 13.7% and 22.4%, from: 14.1% and 22%).

A statistically significant self-reported between-group difference in change was discovered in the use of cars as a primary mean of transportation to and from work. However, it was in an unexpected direction favoring COM (Table 4). Again, more objective data was not analyzed as only 23 (3.3%) employees in EXP and 6 (1.1%) employees in COM had used an accelerometer and a travel diary at both measurement points.

#### 3.2.3. Injuries Due to ACW (Phase 1 and Phase 2)

In Phase 1, 2% (*n* = 18) of the employees reported injuries related to ACW before the environmental improvements and 1.7% did so after the strategies. In Phase 2, the proportion of employees who had experienced ACW-related injuries before social and behavioral strategies was 1% (*n* = 7) in both EXP and COM. After the intervention, the proportions were 0.7% and 0%, respectively.

### 3.3. General Use of the Main Walking and Cycling Path (Phase 1)

At baseline, notably more cyclists (*n* = 646, range 134 to 186) than pedestrians (*n* = 309, range 34 to 133) were observed at four traffic calculation counting points during the afternoon peak hour [24]. After environmental improvements, the corresponding counts were 1013 (ranging from 174 to 333 in the different counting points) in cyclists and 346 (range 47 to116) in pedestrians accounting an overall increase of 36%-points in cyclists and 11%-points in pedestrians.

### 3.4. Quality of the Main Walking and Cycling Path (Phase 1)

Before the environmental improvements, the average speed of cycling was 14 km/h, the comfort of cycling (smoothness of the pavement) 0.97 m/s^2^, and the rate of separation (proportion of separated cycling path from a mixed path) 7% [24]. During the improvements, one traffic light junction was replaced by the right of way junction, but no change in the average speed of cycling was measured. The comfort of cycling improved slightly to 0.88 m/s^2^ and the rate of separation increased from 7% to 70%.

## 4. Discussion

The present study examined the behavioral impact resulting from improved walking and cycling paths. Further, this was followed by workplace-specific social and behavioral strategies on employees ACW in a sample of workplaces in a Finnish urban setting. After improvements, a beneficial pre-post change was found in employees’ intention to cycle at least part of their work journey, but no effects were discovered on actual ACW behavior, albeit the number of pedestrians and cyclists in the improved main path increased in general. Social and behavioral strategies beneficially impacted employees’ willingness to increase walking and cycling in their work journeys and opportunity to cycle at least part of the work journey. No more ACW-related adverse effects were reported after the strategies than before their implementation.

Process evaluation revealed several challenges, which were partly expected in real-world intervention. They affected especially evaluation and are discussed in more detail in the study limitations.

### 4.1. Comparison with Earlier Studies on Environmental Improvements

The studies on building new or improving existing routes show moderate [29,30], contradictory, [31] or no relevant effects [32] on active travel in general. However, only few studies have examined longitudinal impacts specifically on ACW [20]. An intervention conducted in Cambridge, UK included a new guided busway with a path for walking and cycling [33]. The results showed that high exposure to busways increased the proportion of commuting trips involving active travel and decreased the proportion of trips made entirely by car [34]. Furthermore, an intervention carried out in the university campus in Hong Kong included various changes in the built environment (land use, pedestrian network, bus schedules, population density) and was able to increase students’ walking in terms of distance and proportion of trips [35].

Studies comparable to the present study, where the environmental improvements were focused on certain workplace area and the outcome was ACW, were not found in the literature. Thus, direct comparisons with previous findings was not possible. It is likely that the general increase in the proportion of pedestrians and cyclists, which was shown in the main path by the traffic calculations, resulted from improvements made simultaneously to the connecting paths [24]. Research on workplace surroundings may deserve more attention in the future as they have been found an important predictor of mode of travel to work [36].

### 4.2. Comparison with Earlier Studies on Social and Behavioral Strategies

The workplace has been indicated as a potential but under-researched setting for the promotion of active travel [37]. Moreover, perceptions of workplace encouragement has been shown to associate inversely with employees travel mode to work [38]. A recent systematic review included 12 interventions, which were categorized as behavior change programs, workplace travel plans, provision of financial incentives, and introduction of new transport infrastructure [39]. Seven of the studies provided results for physical activity or ACW and six of them found statistically significant effects on either outcome. However, the risk of bias was stated high, especially due to uncontrolled study design and heterogeneity of outcome measures and therefore pooling of the results for meta-analysis was not considered meaningful. Interventions most comparable to the present study were the ones on workplace travel plans, which included research other than individually targeted or infrastructural actions. As only one such study was included [40], no plausible conclusions could have been made. Earlier, Cochrane reviewed the organizational travel plans, which included five workplace studies; they found insufficient evidence for effectiveness [41]. Moreover, these authors call for more robustly designed studies.

Some other multilevel interventions have been reported outside the reviews mentioned above or more recently [42,43,44]. The strategies implemented are quite different, varying from walking champion-led activities [42] to cycling contests, rewards and provision of information [43], and extensive travel plans [44]. Understandably, the results on effectiveness are also inconsistent with two of the studies [43,44], showing small and one [42] to no increase in ACW. To conclude, the findings of the present study about the effectiveness of social and behavioral strategies on employees’ ACW fit into the current equivocal evidence. Well-designed studies with identical contents, outcomes, and measures are needed to get more robust picture of the effectiveness.

### 4.3. Comparison with Earlier Studies on Motivational Factors

Perceptions on work commuting have been shown to differ from those on recreational commuting [20,45], indicating that change may be more difficult to achieve in work than recreational commuting. Only few studies have been published about the impacts of active travel interventions on motivational factors [46] and none of them used similar outcomes to the present study. However, in line with the present findings, willingness to walk or cycle has been seen to increase post-intervention [46]. It has also been shown that attitudes [20] and intentions [47] are powerful contributors to the decision-process related to choosing transport modes. In this respect, the observed positive changes in intentions after environmental improvements in the present study are encouraging. However, as also discovered in the present study, intentions do not necessarily lead to change in actual behavior [47], which is problematic from the promoters’ perspective. More studies are thus needed to close the gap between motivational factors and behavior [47].

### 4.4. Study Strengths

The present study is the first one in Finland, which combines interdisciplinary collaboration between practitioners and researchers working in the fields of transportation, urban design, physical activity, and sustainable development to promote ACW. Previous studies emphasize multisectoral approach in active travel promotion, but at the same time show that coordinated efforts are still scarce [45,48]. Moreover, stakeholders share different perspectives in interpreting evidence and understanding behavior change [49], experiencing a variety of challenges, especially in knowledge exchange [48].

Internationally, the present study is one of the few that has conducted prospective evaluations on promoting active travel in a natural workplace context embedded with randomized controlled design. Most studies look at cross-sectional associations [23], which are unable to provide information on causality of effects. One reason for the small amount of longitudinal real-world studies may lie in pragmatic difficulties in implementation, which is often outside of researchers’ control [50,51]. On the other hand, findings from natural experiments may provide more practice-relevant information to stakeholders and make it more transferrable to non-research contexts [50].

The present study utilizes a socio-ecological framework in ACW promotion in a form, which has not been used in previous studies. The environmental improvements were part of the city traffic plans and the social and behavioral strategies were integrated into the workplaces’ health promotion practices and implemented mostly by the workplaces themselves. Additionally, the intervention included various types of workplaces and the feasibility of the protocol related to the social and behavioral strategies had been previously piloted [52]. All these things are likely to improve the transferability of the results into the practice. 

The study also merits from using multiple outcomes and measures, which increases the likelihood of revealing potential effects. In addition, changes in motivational factors were examined separately from changes in actual ACW behavior, which is suggested in previous literature to gain more detailed information about the intention-behavior gap [47]. Conducting process evaluation improved the external validity of the study increasing understanding about the phenomena behind the behavioral impacts and strengthening transparency of the findings [53]. In addition, adverse effects were assessed after both phases of the study. In a previous systematic review on interventions promoting walking, none of the studies had attempted to do so [54]. Neither were the adverse effects reported in any of the more recent systematic reviews on promoting active travel in general [30] nor specifically ACW in workplace context [39]. However, from the employers’ perspective, safety may play an important role especially when considering engagement in promoting ACW [55].

### 4.5. Study Limitations

The present study has weaknesses, which limit the generalizability and reliability of the results. Firstly, the construction work for main walking and cycling path in Area 1 was markedly delayed, which shrank the exposure time to the path improvements from the intended six to the maximum of two months, depending on the workplace. Similar challenges in construction work have also been faced in previous studies [33,56]. Furthermore, as no parallel strategies were implemented to increase employees’ awareness of the improvements, the exposure time may have been too short for behavioral change, even among the most compliant employees. Previous studies show for example that media campaign may have substantial role in increasing path use [57] and longer exposure periods may enable sleeper, snowball, or threshold effects possibly emerging over time [51].

Secondly, the changes in ACW were based on self-reported information, which may threat the validity of the results due to intentional or unintentional misreporting. The respondents were asked to use accelerometer and travel diary in every measurement point, but contradictory to previous feasibility study in workplaces [58], the number of users was too low for the assessment of changes in both phases of the study. One possible reason for the high accelerometer dropout rates may have been a failure to provide graphical feedback for the employees within a reasonable time. Due to limited staff resources, the time stretched from three weeks to three months, which caused disappointment among users and led to subsequent non-use at the following measurement points. It should also be acknowledged that accelerometers were accompanied with an effortful diary approach in order to separate work trips from overall acceleration data. In recent studies, combination of accelerometer, GPS, and geographical information systems (GIS) data has been shown as a promising method to predict active travel and to decrease the workload for both participants and researchers [59,60].

Thirdly, one half of the workplaces (5/11) in Phase 1 and practically all workplaces (4/5) in Phase 2 went through economic problems and workforce adjustments during the study, which inevitably reduced workplaces’ and employees’ interest towards ACW promotion and study participation. This especially compromised the analysis of change, which required repeated responses from the same participants, and may have led to selection bias and weaker generalizability of the findings, although some earlier studies indicate that individual characteristics have no essential role in incorporating walking and cycling to work journeys [61].

Finally, in Phase 2, most internal teams in EXP named lack of money and staff as major obstacles for implementing actions to promote ACW. As a result, the actions most commonly selected were the ones that were easy and cheap to carry out, such as e-mails and wall posters. Thus, despite of the large number of actions implemented, they may not have been diverse and multilevel enough to bear fruit. It is also possible that after all, actions implemented exclusively at workplace are not sufficient for changing commuting practice, which is highly automated and dependent also on other social contexts [45].

## 5. Conclusions

The current study is a novel example of how to promote ACW within a socio-ecological framework in a real-life setting. It included evaluation of environmental improvements to walking and cycling paths in a longitudinal pre-post design followed by evaluation of workplace-specific social and behavioral strategies in a cluster-randomized controlled condition.

Path improvements in Phase 1 and social and behavioral strategies in Phase 2 facilitated employees’ motivation for ACW but had no impact on their actual behavior, although the general use of the improved main walking and cycling path increased. Nevertheless, observed positive changes in motivational factors are one step forward to behavior change and should be investigated more in future studies. In addition, objective methods, which are feasible in natural experiments with repeated measurements, are warranted for the assessment of ACW.

Several real-world challenges were faced during the study process due to construction work and workforce dynamics, which especially hampered the evaluation. All risks cannot always be anticipated, but acknowledging the most potential ones from the current and previous studies can help to overcome some of them and consequently improve validity and generalizability of the study results.

## Figures and Tables

**Figure 1 ijerph-16-01661-f001:**
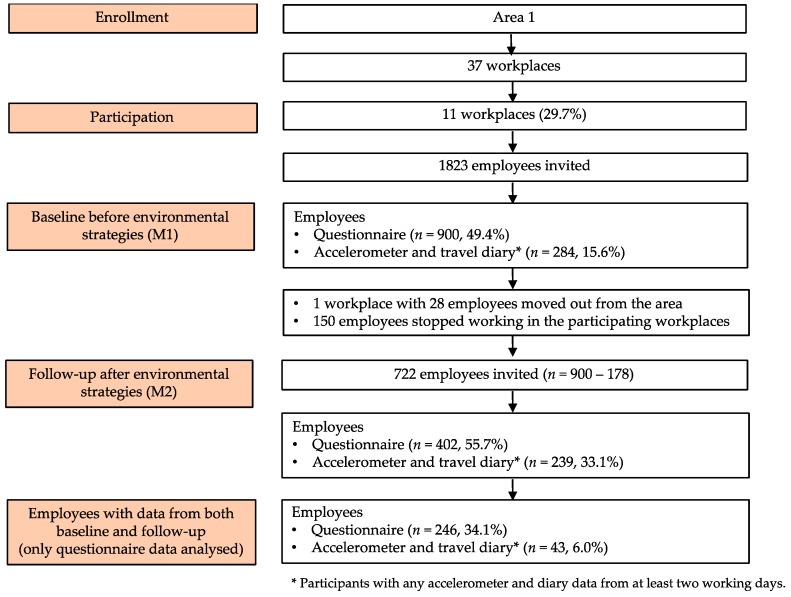
Flow chart of Phase 1 (workplaces in Area 1): Effects of improvements to walking and cycling paths in a longitudinal pre-post design.

**Figure 2 ijerph-16-01661-f002:**
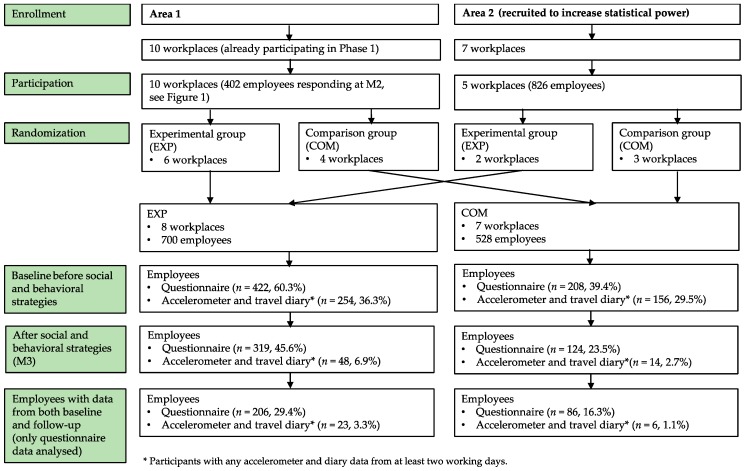
Flow chart of Phase 2 (workplaces in Area 1 and Area 2): Effectiveness of workplace-specific social and behavioral strategies in a cluster-randomized controlled design.

**Table 1 ijerph-16-01661-t001:** Indicators and measures of process and impact evaluation.

Evaluation Questions/Indicators	Measures
**Process evaluation (Phase 1 and Phase 2)**	
What percentage of workplaces volunteered and how representative were they?	Documentation during the recruitment
What percentage of potentially eligible employees took part and how representative were they?	Employee questionnaire
To what extent did the multilevel strategies succeed as intended?	Visual auditing (environmental improvements) Workbooks completed by the workplaces (social and behavioral strategies)
To what extent the employees had made preparations for increasing walking or cycling to work? (Phase 1) perceived effects on their walking and cycling to work? (Phase 1) had acknowledged the strategies implemented? (Phase 2)	Employee questionnaire
**Behavioral impacts (Phase 1 and Phase 2)**	
Motivation for active commuting to work (ACW) Willingness to increase ACW Opportunities to increase ACW Intention to increase ACW	Employee questionnaire
Employees’ ACW Primary mean of transportation to and from work Number of days per week actively commuting the whole journey to and from work Number of days per week walking part of the journey to or from work Number of days per week bicycling part of the journey to or from work	Employee questionnaire, accelerometer and travel diary
Injuries due to ACW Number of employees reporting injuries due to ACW	Employee questionnaire
**General use of the main walking and cycling path (Phase 1)**	
Number of pedestrians and cyclists trespassing the path during the afternoon peak hour	Automatic fixed-point traffic calculations in 4 counting points
**Quality of the main walking and cycling path (Phase 1)**	
Average speed of cycling (km/h) Comfort of cycling = smoothness of the pavement (m/s^2^) Rate of separation = proportion of separated cycling path from a mixed path (%)	Auditing: Cycling with a GPS tracker, which collected information on speed, location and vertical acceleration

**Table 2 ijerph-16-01661-t002:** Baseline characteristics of the participants in Phase 1 and Phase 2. EXP: experimental group; COM: comparison group.

Baseline Characteristic	Phase 1	Phase 2	
	(*n* = 900)	EXP (*n* = 422)	COM (*n* = 208)
Age in years, mean (SD)	43.0 (11.1)	46.7 (10.4)	41.1 (11.2)
Age-group, *n* (%)			
<30 years	123 (13.8)	42 (10.0)	42 (20.4)
3045 years	394 (44.1)	188 (44.7)	88 (42.7)
46–55 years	224 (25.1)	117 (27.8)	49 (23.8)
>55 years	153 (17.1)	74 (17.6)	27 (13.1)
Women, *n* (%)	475 (52.7)	225 (53.3)	107 (51.4)
Body mass index (kg/m^2^), mean (SD)	25.6 (3.9)	26.3 (10.5)	25.6 (4.1)
Body mass index >25, *n* (%)	439 (49.2)	204 (45.2)	100 (49.8)
Smoking; yes, *n* (%)	68 (7.6)	19 (4.6)	14 (6.8)
Married, *n* (%)	714 (79.2)	333 (79.5)	158 (76.3)
Taking care of children <18-years, *n* (%)	387 (43.7)	179 (43.1)	77 (37.4)
Education, *n* (%)			
Secondary school or high school graduate	57 (6.3)	20 (4.7)	9 (4.4)
Polytechnic or vocational school	454 (50.3)	151 (35.8)	91 (44.2)
University degree	388 (43.0)	249 (59.0)	105 (51.0)
Other	4 (0.4)	2 (0.5)	1 (0.5)
Working hours, *n* (%)			
Regular day work	800 (88.9)	364 (90.3)	166 (81.8)
Shift-work (2 or 3 shifts)	32 (3.6)	26 (6.5)	21 (10.3)
Irregular or other hours	11 (1.2)	4 (1.0)	8 (3.9)
Part-time job	34 (3.8)	4 (1.0)	7 (3.4)
Other	23 (2.6)	5 (1.2)	1 (0.5)
Type of work, *n* (%)			
Sedentary work	760 (84.2)	393 (93.8)	162 (78.6)
Mainly standing or light ambulatory work without carrying	79 (8.7)	19 (4.5)	30 (14.6)
Mainly ambulatory work with carrying or climbing stairs	48 (5.3)	6 (1.4)	13 (6.3)
Heavy or extremely heavy physical work	15 (1.7)	1 (0.2)	1 (0.5)
Kilometers (km) from home to work, mean (SD)	14.7 (17.3)	15.3 (21.7)	12.8 (18.2)
<3 km, *n* (%)	77 (8.9)	77 (10.3)	48 (14.0)
3–5 km	132 (15.2)	108 (14.4)	45 (13.2)
5.1–9.9 km	216 (24.9)	195 (26.0)	85 (24.9)
≥10 km	441 (50.9)	369 (49.3)	164 (48.0)

**Table 3 ijerph-16-01661-t003:** Employees’ motivation for Active commuting to work (ACW) before and after Phase 1 and Phase 2. EXP: experimental group; COM: comparison group.

	Phase 1		Phase 2			
	Before	After	Before		**After**	
			**EXP**	**COM**	**EXP**	**COM**
	*n* = 900	*n* = 402	*n* = 422	*n* = 208	*n* = 319	*n* = 124
Willingness to increase walking; yes, *n* (%)	351 (43.2)	160 (43.3)	161 (43.6)	90 (49.2)	115 (38.3)	46 (39.3)
Willingness to increase cycling; yes, *n* (%)	526 (61.7)	223 (57.3)	232 (59.3)	128 (68.1)	185 (59.7)	79 (66.4)
Opportunity of walk at least part of the work journey; yes, *n* (%)	538 (61.4)	231 (57.8)	227 (57.2)	119 (60.4)	201 (63.2)	70 (56.5)
Opportunity of bicycle at least part of the work journey; yes, *n* (%)	565 (65.1)	249 (62.1)	240 (60.2)	134 (68.0)	222 (69.9)	86 (69.4)
Intention to walk at least part of the work journey in the following week, mean number of days (SD)	1.4 (1.8)	2.1 (1.9)	380 (1.8)	180 (2.0)	300 (1.9)	113 (1.8)
Intention to bicycle at least part of the work journey in the following week, mean number of days (SD)	1.7 (1.9)	1.9 (1.7)	386 (1.6)	190 (1.8)	307 (1.9)	118 (1.9)

**Table 4 ijerph-16-01661-t004:** Changes in employees’ motivation for ACW before and after Phase 1 and Phase 2.

	Phase 1 ^1^		Phase 2 ^2^	
	Change (95% CI)	*p*-Value	Between-Group Difference in Change (95% CI)	*p*-Value ^1^
Willingness to increase walking; yes, *n* (%)	3.4% (−3.8 to 10.7)	na	8.7% (1.8 to 15.6)	na
Willingness to increase cycling; yes, *n* (%)	3.9% (−11.0 to 3.1)	na	5.5% (2.2 to 8.8)	na
Opportunity of walk at least part of the work journey; yes, *n* (%)	−3.4% (−9.5 to 2.7)	na	0.5% (−3.5 to 4.5)	na
Opportunity of bicycle at least part of the work journey; yes, *n* (%)	−5.4% (−11.5 to 0.7)	na	5.9% (2.1 to 9.7)	na
Intention to walk at least part of the work journey in the following week, mean number of days (SD)	na	0.50	na	0.06
Intention to *bicycle* at least part of the work journey in the following week, mean number of days (SD)	na	0.001	na	0.13

^1^ Wilcoxon Signed Ranks Test for continuous variables and differences of after-before percentages with 95% Wald interval for difference of proportions with matched pairs for nominal variables. ^2^ Mann-Whitney *U*-test for continuous variables, difference of two independent percentages with 95% confidence intervals for nominal variables. na = not applicable.

**Table 5 ijerph-16-01661-t005:** Employees’ self-reported ACW before and after environmental strategies in Phase 1 and before and after social and behavioral strategies in Phase 2. EXP: experimental group; COM: comparison group.

	Phase 1		Phase 2			
	Before	After	Before		After	
			EXP	COM	EXP	COM
Respondents, *n*	900	402	422	206	319	124
Primary mean of transportation to work, *n* (%)						
By car or motorcycle	543 (60.3)	226 (56.2)	244 (58.1)	100 (48.8)	167 (52.4)	70 (56.0)
By public transportation	131 (14.6)	65 (16.2)	61 (14.5)	29 (14.1)	44 (13.8)	16 (12.8)
By foot	53 (5.9)	32 (8.0)	36 (8.6)	28 (13.7)	35 (11.0)	10 (8.0)
By bicycle	171 (19.0)	74 (18.4)	73 (17.4)	46 (22.4)	71 (22.3)	29 (23.2)
Other	2 (0.2)	5 (1.2)	6 (1.4)	2 (1.0)	2 (0.6)	0 (0.0)
Primary mean of transportation from work, *n* (%)						
By car or motorcycle	535 (59.6)	227 (56.5)	243 (57.7)	101 (49.3)	167 (52.5)	69 (55.2)
By public transportation	129 (14.4)	66 (16.4)	62 (14.7)	30 (14.6)	43 (13.5)	17 (13.6)
By foot	58 (6.5)	31 (7.7)	36 (8.6)	29 (14.1)	34 (10.7)	10 (8.0)
By bicycle	173 (19.3)	74 (18.4)	74 (17.6)	45 (22.0)	72 (22.6)	29 (23.2)
Other	2 (0.2)	4 (1.0)	6 (1.4)	0 (0.0)	2 (0.6)	0 (0.0)
Number of days per week actively commuting the whole journey to and from work, mean (SD)						
Walking	0.0 (1.0)	0.0 (1.0)	0.5 (1.3)	0.8 (1.7)	0.6 (1.5)	0.5 (1.5)
Bicycling	1.0 (2.0)	1.0 (2.0)	0.9 (1.6)	1.1 (1.8)	1.2 (1.8)	1.2 (1.9)
Number of days per week walking part of the journey to work, mean (SD)	1.0 (2.0)	1.0 (2.0)	1.0 (1.8)	1.4 (2.1)	1.2 (2.0)	1.0 (1.9)
Kilometers walked, mean (SD)	0.8 (1.4)	1.0 (1.0)	0.6 (1.1)	0.9 (1.6)	0.6 (1.1)	0.6 (1.3)
Minutes walked, mean (SD)	8.0 (12.0)	6.0 (11.0)	5.5 (9.7)	7.1 (11.7)	6.2 (10.4)	5.4 (9.6)
Number of days per week bicycling part of the journey to work, mean (SD)	1.0 (2.0)	1.0 (2.0)	0.9 (1.7)	1.0 (1.8)	1.1 (1.8)	1.2 (1.9)
Kilometers bicycled, mean (SD)	2.5 (4.1)	2.0 (4.0)	2.1 (4.4)	1.9 (3.7)	2.6 (4.3)	2.5 (5.1)
Minutes bicycled, mean (SD)	9.0 (13.0)	7.0 (13.0)	6.9 (13.4)	6.8 (12.3)	9.1 (13.7)	8.5 (14.7)
Number of days per week walking part of the journey from work, mean (SD)	1.0 (2.0)	1.0 (2.0)	1.1 (1.8)	1.4 (2.1)	1.2 (1.9)	1.1 (1.9)
Kilometers walked, mean (SD)	0.8 (1.4)	1.0 (1.0)	0.6 (1.2)	0.8 (1.4)	0.6 (1.1)	0.7 (1.5)
Minutes walked, mean (SD)	8.0 (13.0)	7.0 (12.0)	6.0 (10.7)	7.4 (12.1)	6.3 (10.7)	6.5 (12.1)
Number of days per week bicycling part of the journey from work, mean (SD)	1.0 (2.0)	1.0 (2.0)	0.9 (1.6)	1.1 (1.8)	1.2 (1.9)	1.2 (1.9)
Kilometers bicycled, mean (SD)	2.5 (4.1)	2.0 (4.0)	2.1 (4.5)	1.9 (3.7)	2.6 (4.3)	2.2 (3.5)
Minutes bicycled, mean (SD)	9.0 (14.0)	7.0 (13.0)	7.1 (13.9)	6.7 (12.4)	9.6 (14.7)	7.8 (11.6)

**Table 6 ijerph-16-01661-t006:** Changes in employees’ self-reported ACW in Phase 1 and Phase 2.

	Phase 1 ^1^		Phase 2 ^2^	
	Change (95% CI)	*p*-Value	Between-Group Difference in Change (95% CI)	*p*-Value
Primary mean of transportation *to* work, *n* (%)				
By car or motorcycle	−1.7 (−6.3 to 3.1)	na	3.1% (0.1 to 6.1)	na
By public transportation	2.1 (-0.8 to 5.0)	na	−1.6% (−4.1 to 0.8)	na
By foot	0.0 (−2.3 to 2.3)	na	2.3% (−0.9 to 5.6)	na
By bicycle	−1.7 (-6.1 to 2.8)	na	−2.3% (−5.5 to 0.9)	na
Other	1.2 (−0.2 to 2.6)	na	−1.5% (na)	na
Primary mean of transportation from work, *n* (%)				
By car or motorcycle	−2.9 (-7.7 to 1.9)	na	3.6% (0.5 to 6.7)	na
By public transportation	3.3 (−3.3 to 6.5)	na	−1.6% (−4.0 to 0.8)	na
By foot	0.0 (−2.6 to 2.6)	na	1.8% (−1.6 to 5.2)	na
By bicycle	−1.2 (-0.6 to 3.3)	na	−2.3% (−5.5 to 0.9)	na
Other	0.8 (−0.3 to 2.0)	na	−1.4% (na)	na
Number of days per week actively commuting the whole journey to and from work, mean (SD)				
Walking	na	0.321	na	0.200
Bicycling	na	0.131	na	0.140
Number of days per week walking part of the journey *to* work, mean (SD)	na	0.499	na	0.120
Kilometers walked, mean (SD)	na	0.456	na	0.749
Minutes walked, mean (SD)	na	0.067	na	0.918
Number of days per week bicycling part of the journey to work, mean (SD)	na	0.505	na	0.502
Kilometers bicycled, mean (SD)	na	0.738	na	0.541
Minutes bicycled, mean (SD)	na	0.395	na	0.908
Number of days per week walking part of the journey from work, mean (SD)	na	0.321	na	0.313
Kilometers walked, mean (SD)	na	0.212	na	0.389
Minutes walked, mean (SD)	na	0.113	na	0.698
Number of days per week bicycling part of the journey from work, mean (SD)		0.627	na	0.586
Kilometers bicycled, mean (SD)	na	0.443	na	0.672
Minutes bicycled, mean (SD)	na	0.224	na	0.960

^1^ Wilcoxon Signed Ranks Test for continuous variables and differences of after-before percentages with 95% Wald interval for difference of proportions with matched pairs for nominal variables. ^2^ Mann-Whitney *U*-test for continuous variables, difference of two independent percentages with 95% confidence intervals for nominal variables. na = not applicable.

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
