# Peer review of "Socio-Ecological Natural Experiment with Randomized Controlled Trial to Promote Active Commuting to Work: Process Evaluation, Behavioral Impacts, and Changes in the Use and Quality of Walking and Cycling Paths"

_ijerph, 2019, doi:10.3390/ijerph16091661_

Round 1

Reviewer 1 Report

Revision of the manuscript entitled "Socio-ecological natural experiment with RCT to promote active commuting to work: process evaluation, behavioural impacts and changes in the use and quality of walking and cycling path".

In the keywords, in my opinion, there are too many words. Perhaps it would be better to delete some of them and only put the most relevant ones.

The introduction is rather short for such a long manuscript. It would be advisable to add more current information on the subject of the research.

The methodology is perfectly detailed.

The results show an important contribution to the scientific community.

The discussion and conclusions are adequate.

Therefore, after these small modifications I recommend its publication.

Author Response

1. In the keywords, in my opinion, there are too many words. Perhaps it would be better to delete some of them and only put the most relevant ones.

We have reduced the number of keywords from 8 to 5 (line 37).

2. The introduction is rather short for such a long manuscript. It would be advisable to add more current information on the subject of the research.

We understand the Reviewer’s concern. However, in this manuscript we have tried to bring a new angle to the topic and to avoid repeating research already brought out in the protocol’s introduction. Also, we have tried to keep the introduction as concise as possible since the manuscript is quite long. However, we have now added information on referred studies in lines 58-65 as requested by Reviewer 3 and highlighted the novelty of our research in lines 74-75 and 424-427 as requested by Reviewer2. Furthermore, as also requested by Reviewer 3, we have added a brief description on socio-ecological approach but inserted it to the methods section (lines 95-100) after the original references (Mc Leroy et al. 1988 and Stokols et al. 1996). More detailed description with a graphical illustration (Figure 2) can be found from the protocol article, which is now also clearly indicated in the text (line 100). 

Reviewer 2 Report

Thank you for the opportunity to review this paper. 

I recommend to extend the introduction with more relevant  and structured information related with your topic. Please, highlight the novelty of your research. 

I recommend to add more 2-3 conclusions to underline your findings.  

Author Response

1. I recommend to extend the introduction with more relevant and structured information related with your topic. Please, highlight the novelty of your research.

We understand the Reviewer’s concern. However, in this manuscript we have tried to bring a new angle to the topic and to avoid repeating research already brought out in the protocol’s introduction. Also, we have tried to keep the introduction as concise as possible since the manuscript is quite long. However, we have now added information on referred studies in lines 58-65 as requested by Reviewer 3 and highlighted the novelty of our research in lines 74-75 and 424-427 as requested by Reviewer 2. Furthermore, as also requested by Reviewer 3, we have added a brief description on socio-ecological approach but inserted it to the methods section (lines 95-100) after the original references (Mc Leroy et al. 1988 and Stokols et al. 1996). More detailed description with a graphical illustration (Figure 2) can be found from the protocol article, which is now also clearly indicated in the text (line100). 

2. I recommend to add more 2-3 conclusions to underline your findings.

We have supplemented the conclusions section with new text, which highlights the novelty of our study and raises issues, which we think that ought to be paid attention in future studies. The new text is in lines 424-427, 430-433 and 435-438. 

Reviewer 3 Report

1.  Line 21: The authors should provide an adequate background of the study.

2.   Line 58-59: More details about these studies should be provided.

3. A socio-ecological approach should be clearly described in the introduction. A key strength of this approach is that it integrates behavior-change strategies at different levels.

4.   Line 74: The title of reference #24 is not found.

5. Authors should make the figure type size large enough to attract the attention of the readers.

Author Response

1. Line 21: The authors should provide an adequate background of the study.

We agree with the Reviewer but also acknowledge the maximum word count for the abstract (200 words). This means that if we increase words to background, we have to omit text from other parts of the abstract, which is challenging considering the complexity of the study design. However, we managed to add one sentence to the background (lines 21-22) and hope that it is now more adequate.     

2. Line 58-59: More details about these studies should be provided.

We have added more details about these studies in lines 58-65.

3. A socio-ecological approach should be clearly described in the introduction. A key strength of this approach is that it integrates behavior-change strategies at different levels. 

We have now added a brief description on socio-ecological approach but inserted it to the methods section (lines 95-100) after the original references (Mc Leroy et al. 1988 and Stokols et al. 1996). More detailed description with a graphical illustration (Figure 2) can be found from the protocol article, which is now also clearly indicated in the text (line 100).

4. Line 74: The title of reference #24 is not found. 

We thank the Reviewer for noticing this. We have added the title of the reference #24 to the reference list.

5. Authors should make the figure type size large enough to attract the attention of the readers.

We have enlarged the font size of the figures from 8 to 10 and changed the font type from Arial to Palatino Linotype, which is the font type in the main text.